# The Discovery of Selective Protein Arginine Methyltransferase 5 Inhibitors in the Management of β-Thalassemia through Computational Methods

**DOI:** 10.3390/molecules29112662

**Published:** 2024-06-04

**Authors:** Bishant Pokharel, Yuvaraj Ravikumar, Lavanyasri Rathinavel, Teera Chewonarin, Monsicha Pongpom, Wachiraporn Tipsuwan, Pimpisid Koonyosying, Somdet Srichairatanakool

**Affiliations:** 1Department of Biochemistry, Faculty of Medicine, Chiang Mai University, Chiang Mai 50200, Thailand; bishant_p@cmu.ac.th (B.P.); ravikumaryuvi@gmail.com (Y.R.); teera.c@cmu.ac.th (T.C.); pimpisid.k@cmu.ac.th (P.K.); 2Department of Biotechnology, Mercy College, Palakkad 678006, India; biotechcreditsu@gmail.com; 3Department of Microbiology, Faculty of Medicine, Chiang Mai University, Chiang Mai 50200, Thailand; monsicha.p@cmu.ac.th; 4Division of Biochemistry, School of Medical Science, University of Phayao, Phayao 5600, Thailand; tipsuwanw@hotmail.com

**Keywords:** β-thalassemia, PRMT5 inhibitors, fetal hemoglobin, molecular docking, dynamics, simulations

## Abstract

β-Thalassemia is an inherited genetic disorder associated with β-globin chain synthesis, which ultimately becomes anemia. Adenosine-2,3-dialdehyde, by inhibiting arginine methyl transferase 5 (PRMT5), can induce fetal hemoglobin (HbF) levels. Hence, the materialization of PRMT5 inhibitors is considered a promising therapy in the management of β-thalassemia. This study conducted a virtual screening of certain compounds similar to 5′-deoxy-5′methyladenosine (3XV) using the PubChem database. The top 10 compounds were chosen based on the best docking scores, while their interactions with the PRMT5 active site were analyzed. Further, the top two compounds demonstrating the lowest binding energy were subjected to drug-likeness analysis and pharmacokinetic property predictions, followed by molecular dynamics simulation studies. Based on the molecular mechanics Poisson–Boltzmann surface area (MM-PBSA) score and molecular interactions, (3R,4S)-2-(6-aminopurin-9-yl)-5-[(4-ethylcyclohexyl)sulfanylmethyl]oxolane-3,4-diol (TOP1) and 2-(6-Aminopurin-9-yl)-5-[(6-aminopurin-9-yl)methylsulfanylmethyl]oxolane-3,4-diol (TOP2) were identified as potential hit compounds, while TOP1 exhibited higher binding affinity and stabler binding capabilities than TOP2 during molecular dynamics simulation (MDS) analysis. Taken together, the outcomes of our study could aid researchers in identifying promising PRMT5 inhibitors. Moreover, further investigations through in vivo and in vitro experiments would unquestionably confirm that this compound could be employed as a therapeutic drug in the management of β-thalassemia.

## 1. Introduction

β-thalassemia is an inherited genetic hemoglobin disorder defined by absent or negligible globin chain synthesis and has become one of the most common disorders worldwide [1,2]. Scientific evidence has suggested that the number of β-thalassemia patients is high in Southeast Asia. According to World Health Organization reports, more than 40,000 infants are carriers of β-thalassemia throughout the world [3]. In Thailand, studies have found that nearly 3250 affected individuals are newly registered every year, and 1 in 180 newly born infants have been identified as β-thalassemia carriers [3]. Clinically, β-thalassemia falls into three main categories: thalassemia major (TM), thalassemia intermedia (TI), and thalassemia minor [4]. TM patients that have presented with acute anemia since childhood periodically require blood transfusions coupled with iron chelation therapy [5,6]. In contrast, thalassemia minor, characterized by a reduced degree of anemia, can be treated with blood transfusions under certain circumstances [5,6]. From a molecular perspective, β-thalassemia is primarily considered a hemoglobinopathy disorder, as the homozygous or heterozygous forms lead to the unbalanced synthesis of α- and β-globin chains. This results in ailing erythropoiesis and the diminished production of regular hemoglobin A [4,7,8]. As a result, the patient can experience severe anemia, pulmonary hypotension symptoms, and pain, all of which necessitate continual medical attention [9,10]. Current treatment regimes, which include iron chelation therapy and blood transfusions, support managing excess iron within the body, but these can also damage certain critical organs like the heart, liver, and endocrine glands [11,12]. Owing to these side effects, a frequent blood transfusion strategy would not be ideal, meaning that the search for an effective alternative for governing ineffective erythropoiesis is of primary importance. Lately, paramount attention has been given to research that focuses on epigenetic regulators and their ability to govern γ-globin expression, as this could help reactivate fetal hemoglobin (HBF) levels [13,14,15].

The importance of understanding the connection of epigenetic regulators in β-thalassemia has progressed exponentially over the last few years. It has been understood that certain key enzymes involved in epigenetic regulation can be exploited to “switch off” β-globin gene synthesis and induce γ-globin expression [16]. In adults, the hemoglobin that carries oxygen is comprised of two γ- and β-globin chains, while in infants, the β-globin is replaced with γ-globin and assembled to form a tetrameric HBF during fetal development [17]. The switching of γ-globin to β-globin gradually occurs several months after childbirth, and this mechanism is termed “globin switching” [16,18]. This shows that a mechanism that can induce HBF synthesis would lead the erythrocytes to prefer γ-globin, which could mitigate the advancement of β-thalassemia [19]. Histone methyltransferases, which belong to a group of histone methyltransferases, catalyze the arginine and lysine present in the histone [20]. S-adenosyl-methionine (SAM), a cofactor, plays a crucial role in this catalysis, and numerous reports have indicated that these enzymes play a crucial role in embryonic development, meiosis, and DNA damage repairs [17]. Importantly, mounting evidence has shown that euchromatic histone lysine methyl transferase 2 (EHMT2), also called G9a, can activate HBF synthesis in adult red blood cells [17,21,22]. Different classes of G9a inhibitors have been developed, and their drug-likeness characteristics have been scrutinized [17,23]. For example, BIX01294, UNC0642, and UNC0638, whose structures are similar to lysine (3(pyrrolidine-1-yl) propoxy group), have all exhibited promising inhibition abilities against G9a [17]. The major drawback of these compounds for clinical use is their poor membrane permeability [17]. Lately, based on the computational approach, N-(4-methoxy-3-methylphenyl)-4,6-diphenylpyrimidine-2-amine and 2-N-[4-methoxy-3-(5-methoxy-3H-indol-2-yl)phenyl]-4-N,6-dimethylpyrimidine-2,4-diamine compounds have been predicted to be potential EHMT2 inhibitors [17,24,25,26]. By exploiting the powerful molecular docking and dynamic simulations strategy, the authors have successfully identified a new class of G9a inhibitors [17]. 

More recently, a liaison between protein arginine methyltransferases (PRMTs) and γ-globin expression has been reported [27]. PRMTs that comprise nine family members have been classified into two types. Type 1 includes PRMT1, 2, 3, 4, 6, and 8, while Type 2 includes PRMT5 and 9 [28,29,30]. Among these, by mediating H4R3me2S (a histone transcriptional repressor), PRMT5 is a suppressor of γ-globin gene expression [30,31,32]. Additional induction of repression is brought on by PRMT5 through assembly of the repressor complex, which consists of histone modifying enzyme SUV4-20h1, casein kinase 2α (CK2α), nucleosome remodelling, and the histone deacetylation complex (NMR) [33] (Figure 1). Having such information, the hunt for potential inhibitors against PRMT5 can be further carried out, and lately, adenosine-2′,3′-dialdehyde has been found to inhibit PRMT5 and would likely be able to reactivate γ-globin gene expression [34]. The outcomes of this study were proven through in vitro testing using human bone marrow erythroid progenitor cells and K562 cells [33,34]. In another study, a tetrahydroisoquinoline-based drug named EPZ015666 exhibited antitumor activity when tested with mantle cell lymphomas [35]. The antitumor activity was found to be based on Smith antigen D3 protein (SmD3) methylation mediated by PRMT5 inhibition [35,36]. Such proven evidence of PRMT5 inhibition using small molecules as inhibitors has spurred researchers to screen and identify more novel compounds that could potentially be used for clinical applications. With an understanding of the role of PRMT5 in managing β-thalassemia well, it is a necessary time to harness computational-guided drug discovery methods to identify new potential PRMT5 inhibitors. This can be accomplished by virtual screening of databases to identify any assuring hit molecules. On the other hand, molecular docking and simulation dynamics have become powerful tools for providing mechanistic insights into determining the stability and affinity of interactions between the virtually screened molecules and the target protein [17,37,38,39]. Nowadays, these computational methods have become a prerequisite because they not only help us to discern the dynamics of the docked compounds with the target proteins but are considered cost-effective, expeditious, and can significantly help in managing late-stage drug failure, as the number of compounds for clinical trials has been restricted to small numbers [17,37,38,39]. With only a few studies having been conducted on the identification of PRMT5 inhibitors using in silico methods, and no previous reports focusing on developing computational-based identification of PRMT5 inhibitors for β-thalassemia, this study examined the adenosine group containing 3XV as a reference compound. A comprehensive virtual screening was conducted in the PubChem database [17]. The compounds that displayed more than 90% structural identity with 3XV were chosen to be further applied to molecular docking and dynamic simulations in order to discover the potential hit compounds that could be employed in developing an ideal drug candidate for managing β-thalassemia [17] via PRMT5 inhibition.

## 2. Results

### 2.1. Virtual Screening and Molecular Docking Analysis

Drug discovery using virtual screening (VS) has become indispensable for quick and profitable drug discovery and optimization. Herein, the crystal structure of PRMT5 (PDB ID 4X61) was selected. The 3XV was taken to screen a compound library obtained from the PubChem database, and the compounds that showed over 90% similarity were filtered out. Around 1934, compounds with high structural similarity were docked into the PRMT5 active site. To validate our molecular docking, the ligand 3XV was docked against PRMT5, and the docked results indicated that the ligand was well bound to the active site and formed similar interactions that were reported in the crystal structure 4X61. The compounds that exhibited better docking scores during the post-docking analysis were ranked in order. The TOP10 ranked compounds that were considered as potential leads were then chosen, and their binding poses and interactions were further analyzed. Autodock Vina 1.1.2 was used to determine the docking scores which ranged from −9.3 to −8.5 kcal/mol (Table 1). When compared to 3XV, (3R,4S)-2-(6-aminopurin-9-yl)-5-[(4-ethylcyclohexyl)sulfanylmethyl]oxolane-3,4-diol (TOP1) and 2-(6-Aminopurin-9-yl)-5-[(6-aminopurin-9-yl)methylsulfanylmethyl]oxolane-3,4-diol (TOP2) displayed the best docking scores at −9.3 and −9.1 kcal/mol, respectively. The residues E444, V503, G438, W579, K333, L319, F327, Y304, F300, F577, F580, S578, and E435 formed a PRMT5 active site within S-Adenosyl-Methionine (SAM), which is essential for substrate methylation (Figure 2). Subsequently, 3XV interacted with PRMT5 by forming hydrogen bonds with F577 and F580. Similarly, the docking results revealed that TOP1 and TOP2 were bound to the active site, forming hydrogen bond interactions with active site residues E444, L437, and W579. Accordingly, Y334 and F577 of PRMT5 also interacted with TOP2 via hydrogen bonds, which indicated that a maximum number of hydrogen bond interactions were formed with the TOP2-PRMT5 complex (Figure 3). The various types of interactions formed by TOP1 and TOP2 have been summarized in Appendix A. The 2D interactions of the TOP10 ranked compounds forming hydrogen bonds, hydrophobics, and other types of molecular interactions with the PRMT5 active site, have been depicted in Appendix A. Apart from these, we also found that docking our compounds did not affect SAM binding. This is evident from the interactions that SAM formed with PRMT5. As shown in Appendix A, the binding of TOP1 and TOP2 in the PRMT5 active site did not affect SAM by forming hydrogen bonds with D419, G365, Y334, Y324, and K333. Following the docking results, the TOP1 and TOP2 compound’s drug properties, pharmacokinetic profiles, and degrees of toxicity were analyzed further.

### 2.2. Pharmacokinetic and ADMET Analysis

In silico validation of novel compounds against PRMT5 is an advantageous method for identifying molecules with high odds of displaying good pharmacokinetic properties. Hence, we have employed web-based software such as Swiss-Adme and ADMET 2.0 to investigate the selected compounds’ drug-likeness capabilities, pharmacokinetics, and toxicological properties. The TOP two compounds with the best docking scores have been selected for the analysis of ADMET properties. First, Lipinski’s rule of five, described as a rule of thumb representing a given compound’s druggability, was applied. Both TOP1 and TOP2 have passed Lipinski’s rule of five (Table 2).

Although the number of hydrogen acceptors for TOP2 was 13, it is known that one violation out of five parameters is generally accepted and does not violate the rule [40,41,42]. The predicted physicochemical properties analyzed through the SwissADME online tool indicate that compounds TOP1 and TOP2 have a log *p*-value of less than five, suggesting that these compounds are likely to exhibit good absorption capabilities. The molecular weight of both compounds was less than 500 Da, while the total polar surface area (TPSA) was around 140 Å (Table 2). Another topological parameter that was checked was the number of rotatable bonds, wherein TOP1 and TOP2 possessed a minimum of one rotatable bond [40,41,42]. ADMET analysis revealed that TOP1 and TOP2 were safe to use in the prediction of the blood-brain barrier (BBB) and P-glycoprotein (P-gp) as negative, which implied that our drug target did not affect the central nervous system and did not lead to multi-drug resistance. Renal clearance, related to a compound’s bioavailability and half-life, is crucial for deciding the dose concentration and dose interval of a chosen drug. Concerning any clinical significance, the total clearance (TC) values of TOP1 and TOP2 were 13.237 and 9.891, respectively. The TC values for both these compounds were 5–15 mL/min/kg, which indicated that the TC values of our compounds were within the moderate range. Apart from the pharmacokinetics, to circumvent the final-step defeat during drug development, the prediction of compound toxicity is of the utmost importance. Our results indicate that TOP1 and TOP2 were negative for AMES and skin toxicity. Breast cancer resistance protein (BCRP), bile salt export pump (BSEP), and hepatic transporters (OATP1B1 and OATP1B3) for TOP1 and TOP2 were also analyzed. Important poly-specific liver transporters (OCT1 and OCT2), which are considered to be crucial for hepatic uptake, were also found to be negative for TOP1 and TOP2. Furthermore, the zero-alert prediction for Acute Toxicity Rule (ATR) signified no adverse effects for TOP1 and TOP2 upon oral and dermal administration. Neither compound exhibited organ toxicity, nor did they display adverse effects on carcinogenicity. Apart from these findings, the other critical factors associated with pharmacokinetic, druggability, and toxicity properties are listed in Table 3. Furthermore, the biological activities of TOP1 and TOP2 were predicted using an online web server (PASS) (https://www.way2drug.com/passonline, accessed on 18 March 2024). This evaluation is crucial as it helps identify the outcome of key biological activities in the human body if a substance is taken as a drug. As has been summarized in Appendix A, the prediction shows that our drug compounds (TOP1 and TOP2) possess antiviral activity against the Hepatitis B Virus, HIV, and the Herpesvirus, while also exhibiting effective antitumor activity against blood cancers in particular. In addition to these findings, other noted activities predicted by the PASS server demonstrate that it can be used as an effective drug in the treatment of various diseases, including liver cirrhosis, myocardial ischemia, metabolic disease, and gout disorders. The other common bioactivities predicted through the use of the PASS server (https://www.way2drug.com/passonline, accessed on 18 March 2024) are listed in Appendix A.

### 2.3. Molecular Dynamics Simulations

After the molecular docking analysis has revealed the best compounds from a vast library, the methods to verify the physical interactions of the docked complexes are of paramount importance. Verification can be conducted by employing MDS, where the interactions between the target compound and the biological macromolecules are investigated under a controlled physiological environment over a defined period. Thus, to confirm the crucial intermolecular interactions and stability of TOP1 and TOP2 with the PRMT5 active site, the simulation was then carried out with the unbound PRMT5 crystal structure and protein-ligand complexes: (i) protein bound with SAM [4X61-APO]; (ii) protein complexed bound with reference compound 3XV [4X61-3XV]; (iii) protein bound with top 2 docking score compounds, TOP1 [4X61-TOP1]; and (iv) protein bound with TOP2 [4X61-TOP2]. These complexes were then subjected to MDS. By analyzing the trajectories for 200 ns time in a solvated medium, the information relevant to molecular interaction dynamics of the desired compounds with the protein could then be unveiled. The molecular interactions formed between the protein-unbound and protein-bound complexes were then investigated based on the trajectories obtained from the root mean square deviations (RMSDs) for all the backbone atoms of the protein and ligand, the root mean square fluctuations (RMSFs) for each single amino acid, the radius of gyration, and the intermolecular hydrogen bond formation.

#### 2.3.1. Root Mean Square Deviation

RMSD is used to evaluate the overall conformational stability and changes in the protein backbone atoms and the docked compound over a defined simulation time. Compared to the crystal structure RMSD, the RMSD values in the protein [4X61-APO] form increased from 0.2 nm to 0.3 nm at 200 ns (Figure 4). When compared with the crystal-bound inhibitor, the RMSD analysis of the TOP1- and TOP2-bound complex revealed that the bound complex was stable over 200 ns but with a higher rise in RMSD values of around 0.3 nm. In the apo form, the RMSD values changed at 84 ns, reached up to 0.4 nm, and remained stable until 100 ns. They then varied at 101 ns, reached 0.5 nm, and stabilized over 200 ns within the 0.3–0.5 nm range. The RMSD in the [4X61-3XV] complex, up to 13 ns, varied at first but stayed constant throughout the experiment. The measured RMSD ranged from 0.3 to 0.4 nm between 13 and 200 ns. The RMSD in the TOP1 complex varied at first at 19 ns, but then stayed constant over the course of the experiment. The measured RMSD ranged from 0.4 to 0.5 nm between 19 and 200 ns. The RMSD values of the TOP2 complex varied at 82 ns, where the difference observed from 0 ns to 80 ns was 0.3 nm but remained stable throughout the simulation within a range of 0.5 to 0.4 nm. This slight perturbation might have been due to the ligand-induced conformational changes, as our targets (both TOP1 and TOP2) had a higher molecular weight and were a bit larger when compared with the crystal-bound inhibitor. Accordingly, we plotted the probability distribution plot as kernel density estimation (KDE) for RMSD distribution. Appendix A illustrates the probability distribution of RMSD in apo and all ligand-bound complexes. The graph shows that 3XV had a broader distribution peak at 0.35 Å RMSD, while the TOP1 and TOP2 distribution readings of the RMSD values were observed between 0.40 to 0.58 Å along with increased density values. Approximately 0.8 nm RMSD was observed for TOP1, and 0.2 nm RMSD was seen for TOP2 based on the probability plots. According to both the RMSD and KDE plots, we can presume that PRMT5-TOP1 deviated less from the crystal structure RMSD than the PRMT5-TOP2 complex.

#### 2.3.2. Root-Man-Square Fluctuation

The root-man-square fluctuation (RMSF) values were determined by measuring overall conformations during a 200 ns simulation to assess the ligand-bound fluctuation of the protein residue movements. From the RMSF trajectories analysis (Figure 5), it is clear that most of the protein residues in the apo form, the known inhibitor, and both TOP1-TOP2-bound complexes had RMSF values below 1.6 Å. Accordingly, APO, 3XV, TOP1, and TOP2 protein complex proteins had average RMSF values of 0.17 ± 0.11 nm, 0.17 ± 0.1 nm, 0.20 ± 0.09 nm, and 0.18 ± 0.14 nm from 0 to 100 ns, respectively. Although there were no significant differences in values across all systems, the RMSF values indicated that the residues in the loop portions of each complex fluctuated less. The noticeable differences in each compound were observed between residues 160 and 180 of Chain B, while TOP2 displayed higher RMSF fluctuations. These residues correspond to the loop region present in the MEP50 domain, which is not involved in SAM or ligand binding. On the other hand, upon investigation, the RMSF of the ligand interacting residues revealed no significant fluctuations observed in each ligand-bound complex, which implied that the local dynamics after each ligand binding were restricted (Appendix A). Notably, the fluctuation patterns were similar despite a slight difference in the G438 residue of the TOP1- and TOP2-bound complex.

#### 2.3.3. Radius of Gyration and Solvent Accessible Surface Area

The rigidness of the apo-protein and the protein-bound complexes for the 200 ns simulation was analyzed by measuring the Rg values. Generally, a correctly folded protein in its native state maintains a stable Rg throughout the simulation period when compared with an unfolded state. With Rg, the relationship between the compactness and flexibility of the protein can be determined by measuring the protein’s solvent-accessible surface area (SASA). To ascertain the effect of inhibitors, we computed the Rg and SASA for apo and each ligand-bound protein system. The kernel density estimated plots (KDE) of Rg and SASA for each system are displayed in Figure 6. The APO KDE figure clearly shows that the enzyme’s Rg value peaked at 2.98 nm with an SASA value of 300.16 Å, while the 3XV KDE plot showed that the Rg value peaked at 3.008 nm with an SASA value of 300.92 Å. Comparably, the Rg value of TOP1 was found to be 3.05 nm and the SASA value was found to be 307.58 Å in the KDE plot. Similarly, the Rg value of TOP2 was found to be 2.99 nm and the SASA value was found to be 295.90 Å. A comparison between TOP1 and TOP2 protein compactness and the solvent-accessible area indicated a very slight deviation in the structure, while TOP1 binding had a more significant effect when compared with TOP2. Overall, it became clear that both the unbound protein and the TOP1- and TOP2-bound complex remained steadily stable throughout the 200 ns, while the measured SASA values implied that the ligand-bound protein complexes did not result in unstable conformation.

#### 2.3.4. Molecular Interactions Analysis

Investigating the molecular interactions between a ligand-bound protein complex is crucial in assessing complex stability during docking and MDS studies. Stability is influenced mainly by the formation of hydrogen bonds and hydrophobic interactions [43,44,45,46], and these interactions between our target compound and PRMT5 were analyzed using the CHAARMM force field. The stability of the ligand-bound protein complex mainly depends on the number of inter-H bonds formed and their interactions. By examining the H-bond occupancy for all the complexes, we found that the maximum number of total H-bonds in both 3XV-bound complexes was 11. The H-bond number slightly decreased in the TOP1-bound protein complex [47], while the lowest was found in the TOP2-protein complex [48]. Despite the differences in the inter-H bonds that were measured, the differences between TOP1 and TOP2 with the reported inhibitor were not quite so high. This shows that our compounds did not induce any significant conformational changes in the protein upon binding. The results of the intermolecular H-bond of the PRMT5-3XV, PRMT5-TOP1, and PRMT5-TOP2 complexes are shown in Figure 7. Following the trajectories, we confirmed that the residues that formed hydrogen bond interactions with respective ligands were similar to the docked complexes. For example, E444 in the active site formed H-bonds with PRMT5-docked with 3XV, TOP1, and TOP2 in docking and MDS trajectories. Similarly, hydrophobic interactions, crucial for ligand binding, were also present in the docked and simulated complexes. As is shown in the figures below, Y304, F327, F577, and F580 were the active site amino acids that formed hydrophobic interactions with TOP1 during docking. The hydrophobic interactions mediated by F327 and F580 were also preserved in the MDS trajectories, which would indicate the stable binding of TOP1 with the PRMT5. Likewise, TOP2, Q309, and F580 helped form H-bonds in the docking and MDS trajectories. In addition to these hydrogen bonds and hydrophobic interactions, π-π stacking and π-cation binding also helped ligand stabilization with the protein complexes. During MDS analysis, we found that π-π stacking interactions were formed in TOP1 and PRMT5-3XV complexes. The molecular contacts established by the active site residues of each ligand-bound complex indicated the stable binding of the ligands with PRMT5.

#### 2.3.5. Principal Component Analysis

Principal Component Analysis (PCA) is a statistical method that helps in the robust analysis of protein conformation clusters and can identify the large and concerted patterns of fluctuations from the MDS trajectories that can be employed to investigate the PRMT5-APO, 3XV, TOP1, and TOP2 complexes. The cumulative movements of the MD trajectories and the conformational changes in PRMT5 ascribed to the binding of the ligands were examined. Conformational analysis of the proteins was used to examine the structural variations amongst the different protein samples. The covariance matrix that was computed once for translational and rotational motions were eliminated from the data. The eigenvectors and eigenvalues were then obtained by diagonalizing the covariance matrices of the backbone C alpha atoms. The collective motions of the C alpha backbone atoms were then recognized as the first two projections (PC1 and PC2) with the highest eigenvalues. Figure 8 displays the 2D plots of PC1 and PC2 for the various protein samples. According to the PCA analysis, the 3XV, TOP1, and TOP2 complexes were stiffer than the apo protein. The plots for these complexes occupy a smaller area and are more densely grouped, as would be expected. Remarkably, the TOP2 complex was not inverted and displayed exceptionally distinct conformations when compared with the other complexes. This implies conformational changes occur from inactive to active structures during TOP2 interactions with PRMT5. These dynamic changes can be correlated with the higher RMSF values observed in PRMT5-TOP2. Using the first two principal components, Gibbs free energy calculations were performed. Appendix A shows the calculated free-energy landscape (FEL) values for all of the complexes. The colour bars indicate the Gibbs free energies ranging from the lowest energy (violet) to the highest energy (dark red) in the conformational states. All the complexes were able to attain the most stable conformations with minimal amounts of energy. In the case of the TOP1-PRMT5 complex, the broader space associated with a lower-energy minima were observed, which was much higher than the TOP2-bound complex. Furthermore, the intermediate minima in the conformational state were higher in TOP2 than in TOP1.

#### 2.3.6. MM-PBSA Analysis

Although the docking score can propose valid knowledge of the appropriate ligand binding pose with the active protein site, the liability between the correlated binding affinities with compound rank order remains uncertain. This unreliability might be because of firm presumptions provided by the scoring functions, which significantly affected the accuracy of the docking score calculations. To overcome this challenge, recently, the exploitation of physical energies (like solvation energy and surface accessibility area through a molecular mechanical force field) has helped to obtain ligand binding energy with promising accuracy. Therefore, the best-docked pose attained during docking studies for each ligand PRMT5-APO, 3XV, TOP1, and TOP2 complexes proceeded for MM-PBSA analysis. Accordingly, the determined binding energy, Van der Waal’s, polar solvation, and electrostatic energy derived from the MDS trajectories, are shown in Table 4. As has been summarized in Table 4, each row indicates a particular system, and the columns indicate different energy components and the degree of binding energy in kJ/mol. The binding energy in the 3XV system was −125.335 ± 19.668 kJ/mol, which suggested a steady association despite a large degree of uncertainty. However, TOP1 exhibited a more prominent binding energy of −148.501 ± 13.847 kJ/mol, suggesting a more robust and stable complex. The binding energy of the TOP2 complex was −130.095 ± 16.505 kJ/mol, indicating a significant degree of stabilization when compared with the solitary 3XV. However, when compared with the TOP1 complex, the TOP2 complex exhibited a less favourable degree of binding energy at −130.095 ± 16.505 kJ/mol, indicating a weaker connection. With a significant binding energy of −148.501 ± 13.847 kJ/mol, the TOP1 complex highlights a solid and stable complex formation. Lastly, the TOP2 complex exhibited a modest binding energy of −130.095 ± 16.505 kJ/mol, suggesting a less prominent but mostly stable degree of interaction. Along with these findings, the electrostatic energy when compared among all the systems, for both TOP1 (−35.603 ± 8.672 kJ/mol) and TOP2 (−55.851 ± 8.957 kJ/mol), exhibited lower electrostatic energy than the 3XV-bound complex (−102.672 ± 11.317 kJ/mol). This shows that both compounds were associated with lower repulsive interactions. Likewise, the positive values observed for all of the complex polar solvation energy outcomes revealed energy gain in the polar solvent. Overall, the binding energy values of TOP1 and TOP2 indicated the stability of the interaction with the PRMT5 active site. Higher negative values indicated better binding capability and more vital complex stability. The differences in the binding energy in these systems might be due to the subtle variations in molecular structure and the type of intermolecular interactions with each ligand.

## 3. Discussion

The primary goal of computational-driven drug discovery is to identify the drug compounds that will empower the efficiency and time duration required during the drug development process. With proven drug potential 3XV, we have chosen 3XV as a lead compound and employed virtual screening of structurally identical compounds in the PubChem library. Furthermore, molecular docking and MDS methods were utilized to identify the promising hit compounds. Approximately 1000 compounds associated with a 95% structural identity with 3XV were retrieved, and molecular docking was employed against PRMT5. The PRMT5, having SAM as a co-factor, revealed an ortho-steric binding site that was surrounded by amino acids viz, E444, F580, F577, F327, Y324, L319, and E435. These amino acids formed H-bond, hydrophobic, π-π stacking, and π-cation binding with the co-crystallized ligand 3XV. Primarily, we first docked 3XV with the PRMT5 active site and found that the 3XV interacts similarly with the active site residue, as can be seen in the reported crystal structure. This was noticeable in the H-bond interaction with 3XV and E444. Similarly, F327 forming π-π stacking interaction was observed in re-docking and was reported with crystal one. These initial findings assured us that docking studies performed via Autodock would be valid, which allowed us to carry out docking for all the compounds obtained from the PubChem database. The compounds having docking scores greater than 3XV were identified from the docked compounds. The docking scores for the selected compounds were between −9.3 and −8.5, where the compounds TOP1 and TOP2 exhibited higher docking scores of −9.3 and −9.1, respectively (Table 1). The docking results ascertained that identical conformations for TOP1 and TOP2 were obtained. Among various docking poses, the pose with more interactions and the H-bond with TOP1 and TOP2 were chosen. As is shown in Appendix A, in the TOP1-docked complex, the oxygen atom (O2) of E444 formed a hydrogen bond with another oxygen atom (O3) of the TOP1. Likewise, in TOP2, the oxygen atom (O2) of F577 produced a hydrogen bond with the nitrogen atom of the ligand, while the nitrogen atom of F80 formed a hydrogen bond with another nitrogen atom of TOP2. Notably, the hydrogen bonds formed by these residues with TOP1 and TOP2 also formed hydrogen bonds with 3XV, as was reported in the crystal form. Finally, regarding the merit of the docking score and the higher molecular interactions, we chose only TOP1 and TOP2 for further MDS studies.

The MDS analysis yielded valid results for the RMSD and RMSF values for all the tested complexes. No considerable instability was observed in the 3XV-PRMT5, TOP1-PRMT5, and TOP2-PRMT5 complexes. All complexes were found to be stable throughout the 200 ns simulation period. While comparing RMSF with residues of TOP1 and TOP2, we found that the residues from 165 to 185 of TOP2 were more flexible than those of the TOP1 and 3XV-bound complexes. Approximately 0.7 nm higher RMSF values were observed than 3XV at residues between 168 and 170. While in the same regions, when compared with the 3XV-bound complex, 0.1 nm of the RMSF results were lower in TOP1. As has become well known from earlier reports, lower RMSF values aid in improving the structural rigidity of the ligand-protein complexes. The RMSF results clearly indicate that TOP1-PRMT5 remains more stable than the TOP2-bound complex. Apart from these outcomes, substantial fluctuations were not observed in other regions of TOP1 and TOP2, but the RMSF values remained the same with 3XV. This would indicate that the secondary structures were well maintained during ligand binding and complex formation. Later, we predicted the binding energy values based on 3XV, TOP1, and TOP2 binding with PRMT5 using MDS-trajectory-derived MM-PBSA outputs. The measured binding energies disclosed in Table 4 were 125.335 ± 19.668 kJ/mol, −148.501 ± 13.847 kJ/mol, and −130.095 ± 16.505 kJ/mol. According to the given binding free energy values, it is presumed that the TOP1 and TOP2 biological activity will be much better. Furthermore, to understand which energy parameters influence the binding of each compound with PRMT5, assessing every energy factor, such as electrostatic, Van der Waal’s, and polar solvation energy, is of paramount importance. These energy values emerge from the interactions formed by the contributions of the active site residues. Van der Waal’s energy contributes to the better binding affinity of TOP1 and TOP2 bound complexes. Similarly, electrostatic energy also favours TOP1 and TOP2 binding with PRMT5. Overall, from the MM-PBSA analysis, it can be inferred that Van der Waal’s electrostatic and polar solvation energy inevitably contributed to TOP1/TOP2 binding. Along with the energies mentioned above, hydrogen bond interactions for TOP1, TOP2, and 3XV binding with PRMT5 are crucial for enduring ligand-bound complex stability. With such given importance, we observed that 3XV built an H-bond with E444 during the docking result analysis. This was verified in the MDS trajectories, where H-bond forming between 3XV and E444 was maintained. Moreover, the H-bond occupancy of the residues that participated in ligand binding of TOP1 and TOP2 was determined. The H-bond formed between 3XV, TOP1, and TOP1 with PRMT5 at 100 and 200 ns was investigated. E444 formed an H-bond with 3XV and continues to be present throughout the MDS trajectories (Figure 9). Likewise, F580 formed a stable H-bond with TOP1 and TOP2 at 200 ns. Notably, in the TOP2-bound complex, Y324 mediates another H-bond with TOP2, as was evident in the MDS analysis. In addition to H-bonds, L436 helps to establish and maintain hydrophobic interactions with the TOP1-bound complex. These molecular interactions prove that TOP1 and TOP2 were well-bounded to the PRMT5 active site and could remain stable as complexes for a definite period to inhibit enzyme activity, which was demonstrated by H-bond occupancy (Appendix A). Furthermore, the structural snapshots of each ligand-bound complex observed at 0, 50, 100, 150, and 200 ns revealed that all ligands 3XV, TOP1, and TOP2 were determined to be intact in the active site throughout the simulation period. Interestingly, in the TOP1-PRMT5 complex, after 100 ns, the position of SAM was slightly moved from its original location (Appendix A). Nevertheless, this SAM displacement did not affect the TOP2 binding action with the protein, while there were no distortions in the overall structure. Meanwhile, the displacement of SAM from its original active site also hampered the enzyme activity as it required the active site amino acid residues for methylation. This would make TOP1 an efficient inhibitor to suppress PRMT5 activity. Overall, this study helped us to identify the two promising novel PRMT5 inhibitors based on the well-known PRMT inhibitor, 5′-deoxy-5′methyladenosine (3XV) ligand structure. Remarkably, TOP1 displayed solid potential to be developed as a drug molecule in managing β-thalassemia because, apart from covalent binding, non-covalent interactions, such as those involving π-π stacking, were found to be formed by F327 and F80 with TOP1. In general, π-π stacking and cation-π interactions were found to have played a major role in the pharmacology and function of different membrane proteins and ion channel receptors. For example, from earlier reports, it has become evident that cation-π interaction is essential for Ach receptors to bind with nicotine in the brain. Taken together with the results provided in this study, future studies, including those employing in vitro methods, need to be performed to validate the potential hit compounds identified in this study.

## 4. Materials and Methods

### 4.1. Virtual Screening

The PubChem database containing around 96 million compounds was harnessed to perform virtual screening and to identify the promising ligands that could be PRMT5 inhibitors [49]. A similarity search was conducted to identify the compounds that exhibited higher similarity with the ligand 3XV. The PubChem database, which uses the Tanimoto-based 2D fingerprint similarity search approach, generated a library of 985 hits with >90% similarity. These compound structures were saved in Structure-Data File (SDF) format and used for further analysis. Pymol, Biovia Discovery Studio, and Maestro were used for protein visualization and for preparing graphical representations.

### 4.2. Protein Preparation and Molecular Docking

The human PRMT5 domain crystal structure complex with inhibitor 3XV (PDB ID: 4X61) was downloaded from the RCSB Protein Data Bank [25], and protein preparation was completed using Biovia Discovery Studio [50]. This enabled us to prepare the protein that was most applicable for further docking. First, the co-crystallized ligand 3XV was removed manually from PRMT5, followed by deletion of water molecules and heteroatoms. SAM was not removed from the protein. Furthermore, hydrogen atoms were introduced, bond orders were assigned, and missing atoms and residues, if present, were filled. Next, the His, Asp, and Glu residues were protonated as has been previously described to establish the hydrogen bond network during the docking phase. AutoDock Vina was utilized for molecular docking, and the docking protocol was established as per the previous study, where the search space and grid box were defined, and the docking runs were initiated [17]. Once the docking was completed, the docking poses and their socking scores were analyzed and ranked according to each ligand binding affinity kcal/mol value. Software programs like PyMol 2.5.5 and Discovery studio visualizer 2021 were used to demonstrate the binding interactions of the 3XV and the top 10 compounds with the PRMT5 active site [50,51].

### 4.3. Physiochemical and Drug-Likeness Properties 

The physicochemical properties of 3XV, TOP1, and TOP2 were analyzed using Swiss-ADME software (http://www.swissadme.ch/, accessed on 9 December 2023). The drug-likeness properties, such as molecular weight, H-bod donors, H-bond acceptors, logP Values, and other critical criteria, were assessed as per Lipinski’s rule of five (http://www.swissadme.ch/, accessed on 9 December 2023). Accordingly, ADMET 2.0 was used to determine certain parameters, such as toxicity, metabolism using cytochrome P450, gastrointestinal absorption, and excretion (https://admetmesh.scbdd.com/, accessed on 8 December 2023).

### 4.4. Molecular Dynamics

The top two compounds—reference compound (3XV) and crystal structure of PRMT5—were selected for Molecular Dynamics (MD) simulation. The MD simulation was conducted using GROMACS version package-2019.4 (http://www.gromacs.org/, accessed on 9 December 2023) and a CHARMM36 force field [52]. The input generator’s solution builder feature was utilized in the construction of each complex. To solvate each system, a rectangular water box was employed with a 10 Å separation between its edges, using the TIP3 model. The topologies of all systems were analyzed using the CHARMM36 force field, and coordinate files were generated accordingly. Molecular dynamics simulations (MDS) were carried out under physiological conditions of 310 K and neutral pH, using GROMACS v2019.4 simulation software. Hydrogens were added to heavy atoms, and protein coordinates were converted using pdbgmx2. Subsequently, TIP3 water molecules were replaced with sodium and chloride ions to maintain salt concentration, achieved through the genion plugin in GROMACS to neutralize water atoms. Convergence criteria with Emtol were applied to limit bond lengths to a maximum force of 1000 kJ/mol while the steepest descent technique was used to maintain a constant temperature of 310 K over 50,000 steps. A cubic box with water extending 10 Å around the protein was used to solvate the structures, employing a TIP3 water model. Energy minimization focused on neutralization to eliminate irregular torsions, and electrostatic interactions were computed using the Particle Mesh Ewald (PME) method. Further equilibration of the system involved progressively removing constraints on heavy atoms using the number of atoms, volume, and temperature (NVT), as well as pressure and temperature (NPT). Conformational adjustments were made to refine the processing system for a duration of 200 ns in MDS. Solvation was achieved through the simple point charge water method. The ligands and proteins were merged to prepare the ligand-protein complexes. A box of cuboid shape was generated and positioned to fit the ligand-protein complex within it. Na^+^ and Cl^−^ ions were used to maintain the system electroneutrality. Furthermore, the apo-protein content and ligand-bound protein energy minimization of the complex were then determined by employing the steepest descent minimization algorithm and the outcomes were then further equilibrated with NVT and NPT simulations. The resulting complex was employed for 200 ns simulation time. RMSD was employed to investigate the structural stability of the apo protein and the ligand-protein complexes. Flexibility examination by RMSF, compactness through Rg, solvent-accessible surface determination by SASA, hydrogen-bond analysis, and principal component analysis (PCA) were studied using GROMACAS commands. Accordingly, Xmgrace was employed to visualize the 2D plot of the analysis mentioned above.

### 4.5. Free Energy Landscape (FEL) and MM-PBSA Analysis

The minimum states of the apo protein and ligand-bound protein complexes were determined using free energy landscape analysis (FEL). The ‘gmx sham’ GROMACS command was employed for FEL calculation. A Python script was utilized to visualize and generate the 3D images. For the binding of free energy predictions, the g_mmpbsa tool was employed [53]. The final 50 ns of simulation trajectories were taken for the ΔG calculation. The energy components were evaluated to calculate the binding affinity and the involvement of multiple energy terms contributing to the overall binding energy. The ΔG was then calculated by applying the following equation:ΔG_binding_ = G_complex_ − (G_protein_ + G_ligand_)(1)
where G complex denotes the free energy of the protein-ligand complex (PL, where ligand usually will be a substrate or inhibitor) and the G_protein_ and G_ligand_ corresponds to the free energy values of the ligand and protein.

The free energy in the complex and apo forms was calculated by employing the following equation:G_x_ = (E_MM_) − TS + (G_solv_)(2)
where X denotes the PL complex or apo form, namely P or L, the average molecular mechanics was determined by energy molecular mechanics (E_MM_), temperature change in entropy (TS) denotes the entropic contribution, and G_solv_ refers to the free energy solvation of ligand binding with the protein.

The molecular mechanics (E_MM_) were analyzed using the electrostatic and van der Waal’s (bonded and non-bonded) interactions between ligand and protein, as has been depicted in Equation (3). G_solv_ represents the linear Poisson Boltzmann equation for individual states (G_polar_), and non-hydrophobic interactions were taken to determine the solvent-accessible surface area values.
E_MM_ = E_bonded_ + E_non-bonded_(3)
G_solv_ = G_nonpolar_ + G_polar_(4)

## 5. Conclusions

This study explored a virtual screening method employed to identify new compounds that could be used as potential PRMT5 inhibitors. A compound library retrieved from the PubChem database based on the co-crystallized PRMT5 inhibitor was employed for docking studies to achieve this goal. The identified compounds with the highest binding affinity were listed with the help of an auto dock score lower than 15 kcal/mol. The top two ranked compounds approved the false-positive evaluation during PAINS analysis and passed the required ADMET properties. Furthermore, the docking simulations and MM-PBSA analysis indicated that TOP1 and TOP2 exhibited higher degrees of binding affinity when compared with 3XV. Afterward, the best-docked pose for 3XV, TOP1, and TOP2 complexed with PRMT5 was employed in 200 ns simulations to unfold the binding strength and molecular interaction stability and to determine the novel interactions seen in the docked complex. Subsequently, based on the RMSD, RMSF, Rg analysis, and intermolecular interactions of each ligand complex in MD trajectories, TOP1 exhibited stable binding with the PRMT5 active site. In summary, the TOP1 compound exhibited promising potential and can be developed as an ideal drug candidate for β-thalassemia when proven with in vitro and in vivo experimental results.

## Figures and Tables

**Figure 1 molecules-29-02662-f001:**
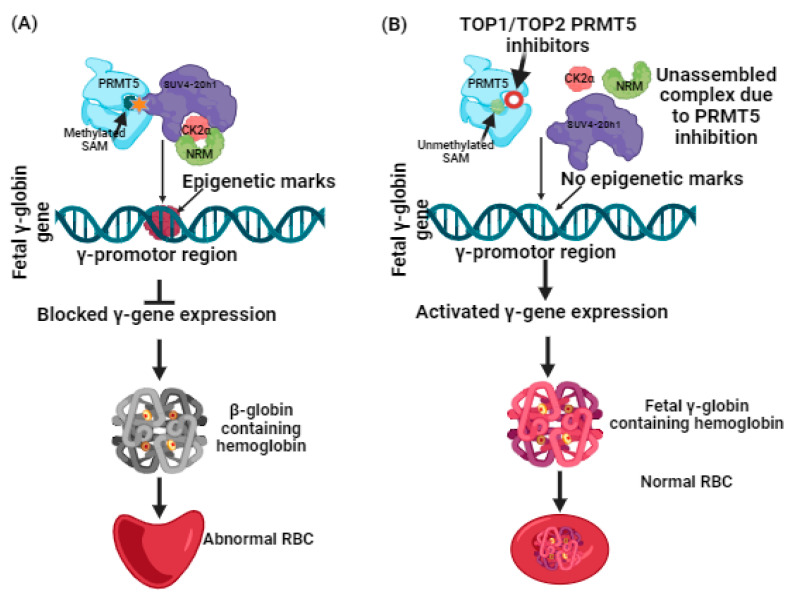
(**A**) Suppression of γ-globin gene expression by PRMT5 and its complexes. (**B**) Reactivation of γ-globin gene expression by inhibiting PRMT5 leads to γ-globin-containing hemoglobin synthesis.

**Figure 2 molecules-29-02662-f002:**
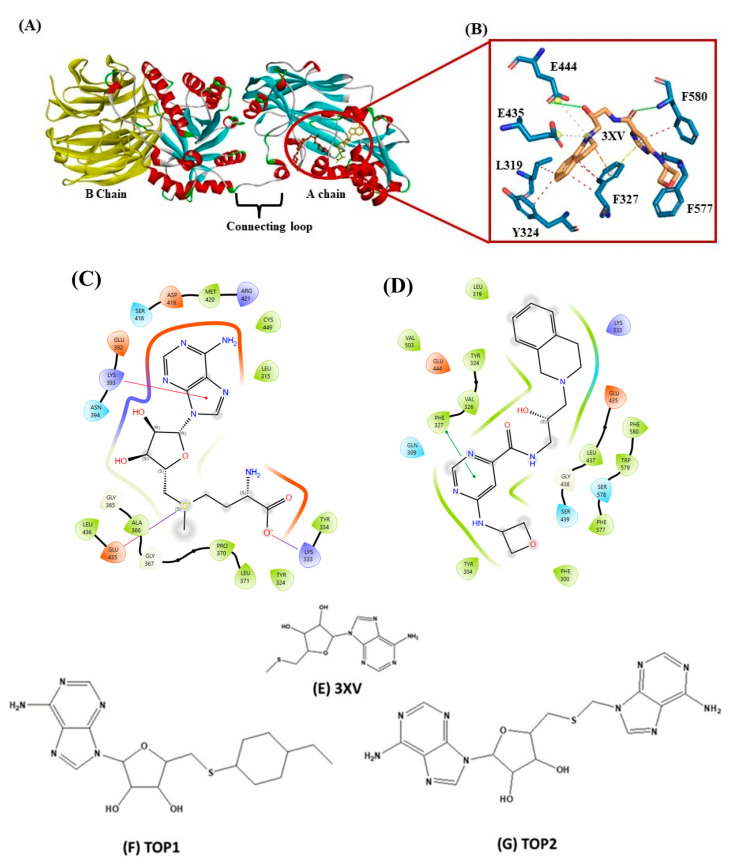
(**A**) Cartoon representation of crystal structure of PRMT5. (**B**) Three-dimensional stick representation of PRMT5 active site docked to 3XV ligand. (**C**) Two-dimensional representation of SAM binding site. (**D**) Two-dimensional representation of 3XV binding site. (**E**) Two-dimensional structure of 3XV. (**F**) Two-dimensional structure of TOP1. (**G**) Two-dimensional structure of TOP2.

**Figure 3 molecules-29-02662-f003:**
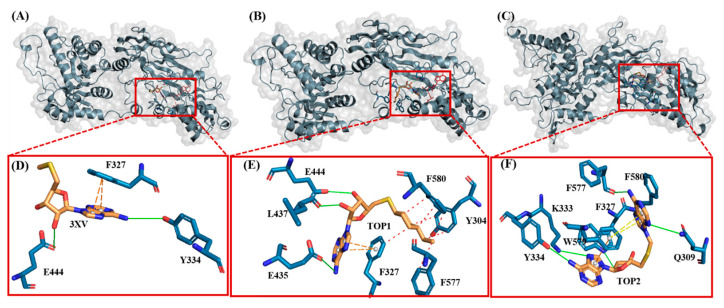
Illustration of PRMT5-docked with different ligands. In (**D**–**F**), active site residues forming interactions represented by blue sticks, ligands represented by orange sticks. Interactions represented by straight lines and dotted lines (green-hydrogen bond, blue-hydrophobic, π-π stacking-orange. (**A**) PRMT5-docked with 3XV. (**B**) PRMT5-docked with TOP1. (**C**) PRMT5-docked with TOP2. (**D**) Three-dimensional stick representation of PRMT5 active site bound to 3XV. (**E**) Three-dimensional stick representation of PRMT5 active site bound to TOP1 (**F**) Three-dimensional stick representation of PRMT5 active site bound to TOP2.

**Figure 4 molecules-29-02662-f004:**
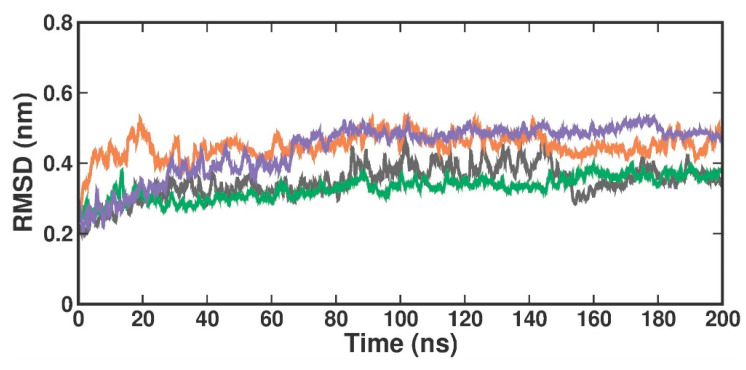
RMSD plot of docked complexes generated through MDS at 200 ns. Gray—RMSD conformational dynamics analysis of APO. Green—RMSD conformational dynamics analysis of 3XV. Orange—RMSD conformational dynamics analysis of TOP1. Violet—RMSD conformational dynamics analysis of TOP2.

**Figure 5 molecules-29-02662-f005:**
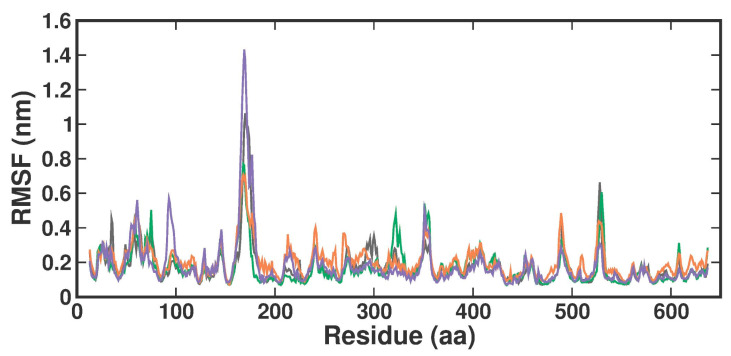
RMSF plot of docked complexes generated through MDS at 200 ns. Gray—RMSF conformational dynamics analysis of APO. Green—RMSF conformational dynamics analysis of 3XV. Orange—RMSF conformational dynamics analysis of TOP1. Violet—RMSF conformational dynamics analysis of TOP2.

**Figure 6 molecules-29-02662-f006:**
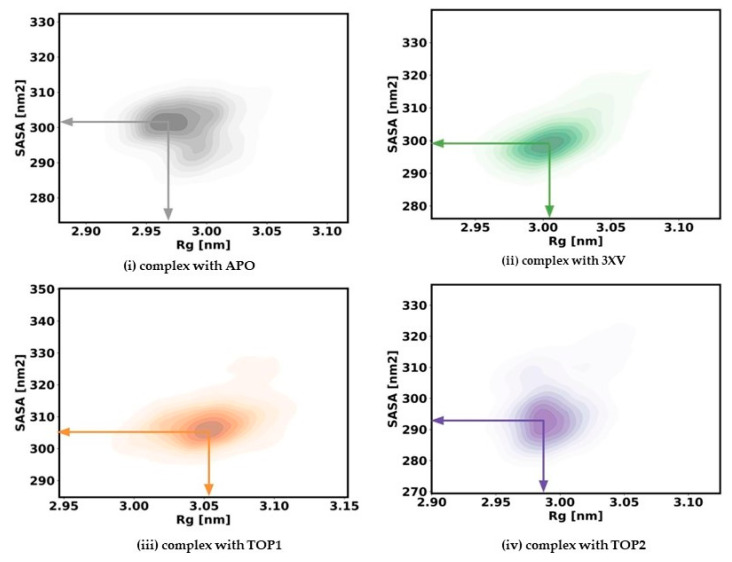
Rg and SASA plots of docked complexes generated through MDS at 200 ns. Gray—Rg plot of PRMT5 in apo and complexed with APO. Green–Rg plot of PRMT5 in apo and complexed with 3XV. Orange—Rg plot of PRMT5 in apo and complexed with TOP1. Violet—Rg plot of PRMT5 in apo and complexed with TOP2.

**Figure 7 molecules-29-02662-f007:**
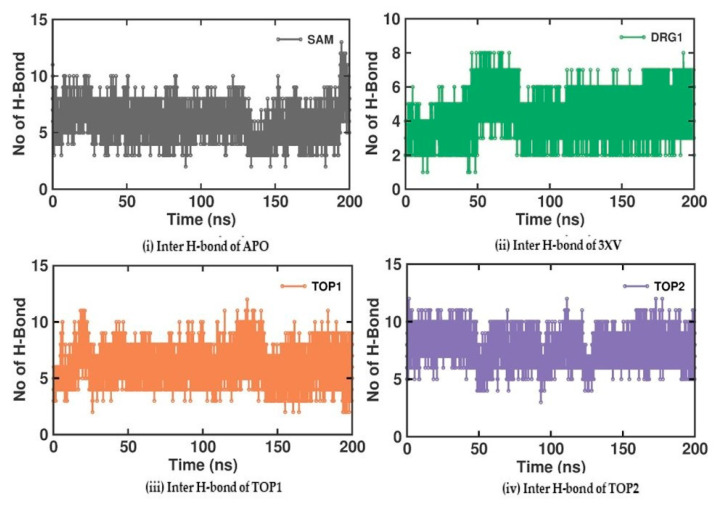
H-bond analysis of docked complexes generated through MDS at 200 ns. Gray—Inter H-bond analysis of APO. Green—Inter H-bond analysis of 3XV. Orange—Inter H-bond analysis of TOP1. Violet—Inter H-bond analysis of TOP2.

**Figure 8 molecules-29-02662-f008:**
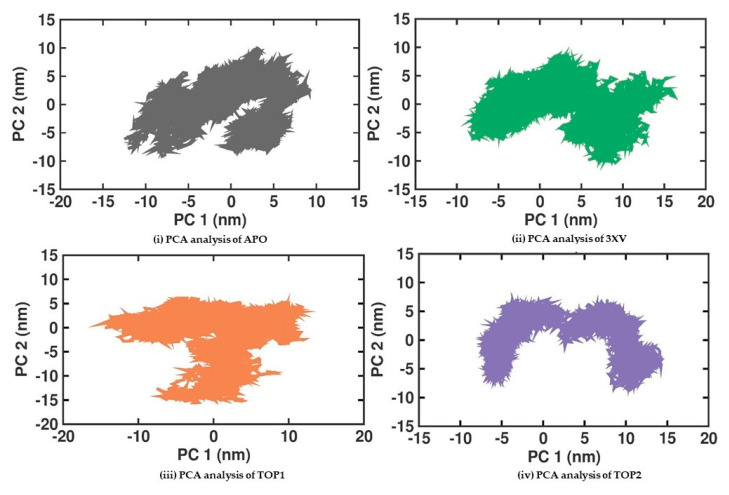
PCA analysis of docked complexes generated through MDS at 200 ns. Gray—PCA analysis of APO. Green—PCA analysis of 3XV. Orange—PCA analysis of TOP1. Violet—PCA analysis of TOP2.

**Figure 9 molecules-29-02662-f009:**
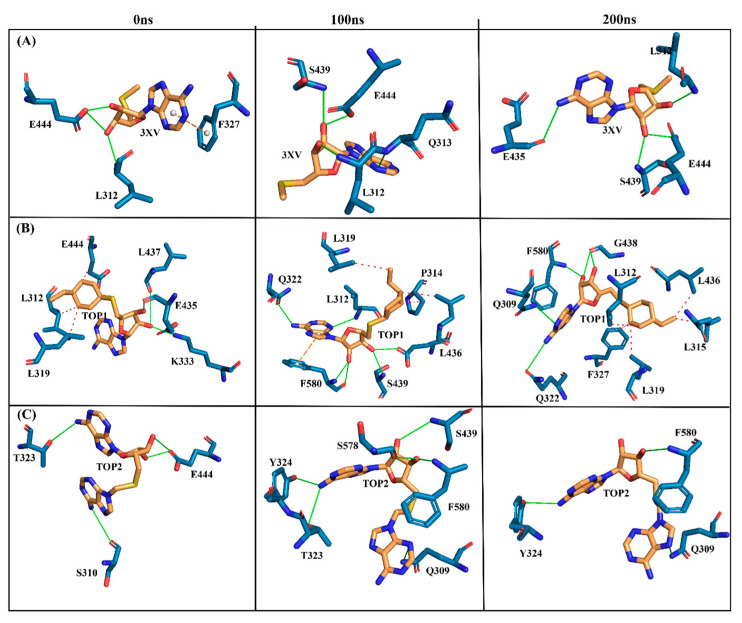
Three-dimensional stick representations of simulated docked complexes at different time intervals. (**A**) 3XV. (**B**) TOP1. (**C**) TOP2. Green line represents H-bond interaction. Red line represents hydrophobic interaction. Orange line represents π-π- stacking interaction. Yellow line represents cation-π stacking.

**Table 1 molecules-29-02662-t001:** Compound name, PubChem ID, and top 10 IFD scores were obtained during docking studies.

Ligand	Code	PubChem ID	Autodock Score (Kcal/mol)
5′-Deoxy-5′-methylthioadenosine	3XV	439176	−6.6
(3R,4S)-2-(6-aminopurin-9-yl)-5-[(4-ethylcyclohexyl)sulfanylmethyl]oxolane-3,4-diol	TOP1	142801950	−9.3
2-(6-Aminopurin-9-yl)-5-[(6-aminopurin-9-yl)methylsulfanylmethyl]oxolane-3,4-diol	TOP2	134460148	−9.1
(2R,3R,4S,5S)-2-(6-aminopurin-9-yl)-5-(2-cyclohexylethylsulfanylmethyl)oxolane-3,4-diol	TOP3	68593553	−9.0
4-[[(2S,3S,4R,5R)-5-(6-aminopurin-9-yl)-3,4-dihydroxyoxolan-2-yl]methylsulfanylmethyl]-5-hydroxy-1,3-dihydroimidazol-2-one	TOP4	90743419	−8.8
2-[1-(2-Amino-ethyl)-piperidin-4-ylsulfanylmethyl]-5-(6-amino-purin-9-yl)-tetrahydro-furan-3,4-diol	TOP5	67994427	−8.7
(2R,3R,4S,5S)-2-(6-aminopurin-9-yl)-5-(cyclohexylsulfanylmethyl)oxolane-3,4-diol	TOP6	44401867	−8.6
2-(6-Aminopurin-9-yl)-5-(heptylsulfanylmethyl)oxolane-3,4-diol	TOP7	360632	−8.6
(3R,4S,5S)-2-(6-aminopurin-9-yl)-5-[(3-chlorocyclohexyl)sulfanylmethyl]oxolane-3,4-diol	TOP8	142801994	−8.6
2-(6-Aminopurin-9-yl)-5-(piperidin-4-ylsulfanylmethyl)oxolane-3,4-diol	TOP9	56671175	−8.5
(2R,3R,4S,5R)-2-(6-aminopurin-9-yl)-5-(2-cyclohexylsulfanylethyl)oxolane-3,4-diol	TOP10	163984180	−8.5

**Table 2 molecules-29-02662-t002:** Drug-likeness properties of selected compounds obtained using Swiss-ADME.

Ligand (PubChem ID)	cLogP	Solubility	Mol wt	TPSA	nHA	nHD	nROT	Drug Likeness	Drug-Score
439176-3XV	−0.66	−3.21	297.0	144.6	8	4	3	−6.21	0.45
142801950-TOP1	1.69	−4.96	393.0	144.6	8	4	5	−5.21	0.33
134460148-TOP2	−0.69	−1.33	430.0	214.2	13	6	5	−5.56	0.42

**Table 3 molecules-29-02662-t003:** Pharmacokinetic properties of selected compounds obtained using ADMET2 and admetSAR.

	Activity	3XV	TOP1	TOP2
A (Absorption)	Human intestinal absorption (HIA)	Positive	Positive	Positive
	Human oral bioavailability (HOB)	Negative	Negative	Negative
	Caco-2 permeability	Negative	Negative	Negative
D (Distribution)	Plasma protein binding (PPB)	0.369297683	0.358286798	0.103410304
	P-glycoprotein substrate, inhibitor:			
	Substrate	Negative	Negative	Negative
	Inhibitor	Negative	Negative	Negative
	Blood-brain barrier penetration (BBB)	Positive	Positive	Positive
M (Metabolism)	Cytochrome P450 (CYP450) substrate, inhibitor:			
	Substrate:			
	✓ CYP2C9	Negative	Negative	Negative
	✓ CYP2D6	Negative	Negative	Negative
	✓ CYP3A4	Negative	Positive	Negative
	Inhibitor:			
	✓ CYP1A2	Negative	Negative	Negative
	✓ CYP2D6	Negative	Negative	Negative
	✓ CYP2C9	Negative	Negative	Negative
	✓ CYP2C19	Negative	Negative	Negative
	✓ CYP3A4	Negative	Negative	Negative
	Pharmacokinetics transporters:			
	BRCPi	Negative	Negative	Negative
	BSEPi	Negative	Negative	Negative
	OCT1i	Negative	Negative	Negative
	OCT2i	Negative	Negative	Negative
	MATE1i	Negative	Negative	Negative
	OATP1b1i	Positive	Positive	Positive
	OATP1b3i	Positive	Positive	Positive
	OATP2b1i	Negative	Negative	Negative
E (Excretion)	Renal clearance	−6.21	−5.21	−5.56
T (Toxicity)	Organ toxicity:			
	Drug-induced liver injury	Negative	Negative	Negative
	Human either-a-go-go-related gene (hERG) inhibition	Negative	Negative	Negative
	Acute toxicity	1.401126266	2.040514946	1.306819558
	Eye injury	Negative	Negative	Negative
	Eye corrosion	Negative	Negative	Negative
	Genomic toxicity			
	Ames mutagenesis	Negative	Negative	Negative
	Carcinogenesis	Negative	Negative	Negative

**Table 4 molecules-29-02662-t004:** Different energy calculation predictions were observed in the following systems: PRMT5-3XV, PRMT5-TOP1, and PRMT5-TOP2.

Ligand (PubChem ID)	Van der Waal’s Energy (kJ/mol)	Electrostatic Energy (kJ/mol)	Polar Solvation Energy (kJ/mol)	SASA Energy (kJ/mol)	Binding Energy (kJ/mol)
439176 (5′-Deoxy-5′-methylthioadenosine/3XV)	−157.100 ± 13.755	−102.672 ± 11.317	151.090 ± 19.812	−16.653 ± 1.103	−125.335 ± 19.668
142801950 ((3R,4S)-2-(6-aminopurin-9-yl)-5-[(4-ethylcyclohexyl)sulfanylmethyl]oxolane-3,4-diol/TOP1)	−203.684 ± 6.001	−55.851 ± 8.950	149.935 ± 19.952	−20.494 ± 0.872	−130.095 ± 16.505
134460148 (2-(6-Aminopurin-9-yl)-5-[(6-aminopurin-9-yl)methylsulfanylmethyl]oxolane-3,4-diol/TOP2)	−230.417 ± 11.119	−35.603 ± 8.672	140.677 ± 19.708	−23.157 ± 1.075	−148.501 ± 13.847

## Data Availability

All data presented in this study are available from the corresponding author upon reasonable request.

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
