# Peer review of "The Discovery of Selective Protein Arginine Methyltransferase 5 Inhibitors in the Management of β-Thalassemia through Computational Methods"

_molecules, 2024, doi:10.3390/molecules29112662_

Round 1
Reviewer 1 Report (New Reviewer)
Comments and Suggestions for Authors
To the authors,
This work by Pokharel et al. summarizes the discovery of selective protein arginine methyltransferase 5 inhibitors for managing β-thalassemia via virtual screening, docking and molecular dynamics. The manuscript has been well written with the authors observations and conclusions drawn thereof seem to be in agreement. However, there are some minor issues that are as follows:
1. It would perhaps be better if the authors show a figure with structures of 3XV, Top1 and top2 for better clarity in addition to the molecular representations.
2. In table 4, it would help if the authors mention the compounds in parentheses after there pubchem Id. In the same table, there is a formatting error for the electrostatic energy of the compound 142801950, where all other values are in 3 decimal places versus 2 for this solitary value.
3. For the MM-PBSA analysis, given the strong dependence on initial conformations, it is strongly recommended to start with top 3 poses for each compound, perform equivalent MD and then MM-PBSA to get robust estimates for each of the individual compounds.
Author Response
Please see the attachment

Reviewer 2 Report (New Reviewer)
Comments and Suggestions for Authors
The authors employed the standard protocol of computer-aided drug design to discover new PRMT5 inhibitors. All data are clearly presented and discussed. The results seem to be solid, although it is always desired to have experiments to support the results. I have only a few comments about the writing:
1. Many abbreviations are used without explanation. For instance, MM-PBSA and MDS (page 1), SmD3 (page 3), TC values (page 7), FEL (page 13), EMM and TS (page 19). These terms may be explained later, but the tradition is to give the full spelling the first time the term appears.
2. Citations for the tools/methods used: PASS, Swiss-Adam, ADMET2.0, GROMACS, CHARMM36, g_mmgbsa, etc. should be cited.
3. Gromacs job setup should be explained instead of just citing their own paper.
4. All figures use color for labeling. Especially Fig. 6, 7, 8. It is recommended to label the panels individually, as some readers may be color blinded.
5. Page 13, line 381-382: This implies conformational changes occur from inactive to active structures during TOP2 interaction with PRMT5. Did the authors observe this conformational change during MD simulations? Can the authors simply show the protein conformations to support the statement?
6. Labels of supplementary material sometimes do not match. For instance, Page 13 line 385 says “Figure S4 shows the calculated FEL values for all the complexes.” But actually, free energy landscapes are depicted in Figure S5. Same for Figure S5, S6 on page 17. There may be more contents mislabeled. The authors should do a final check on the labels of contents in the supplementary material.
Round 2
Reviewer 1 Report (New Reviewer)
Comments and Suggestions for Authors
The revised manuscript represents much warranted changes as compared to the initially submitted version. There were some minor typos but it can be fixed by a thorough proofreading.
This manuscript is a resubmission of an earlier submission. The following is a list of the peer review reports and author responses from that submission.
Round 1
Reviewer 1 Report
Comments and Suggestions for Authors
The manuscript entitled “Discovery of selective protein arginine methyltransferase 5 inhibitors for managing β-thalassemia through computational methods” discloses the computational discovery of PRMT5 inhibitors as promising lead compounds that could be considered for the development of new drugs for the treatment of β-thalassemia. The chosen target is of interest and the results are novel and comprehensive. However, extensive language check is necessary especially in the abstract, introduction and discussion part of the manuscript.
The introduction part is well documented; however, it needs extensive language check.
The research presented is comprehensive utilization of legitimate and reliable computational methods used in the drug discovery, such as virtual screening of compound databases, molecular docking and dynamics, statistical analysis etc. Resulting in two new compounds with computationally predicted inhibitory activity towards PRMT5. No new software was disclosed. Results are well discussed, although improvements could be made in the discussion part.
I suggest publishing after language correction.
Comments on the Quality of English Language
Extensive language check is necessary especially in the abstract, introduction and discussion part of the manuscript.
Reviewer 2 Report
Comments and Suggestions for Authors
In "Discovery of selective protein arginine methyltransferase 5 in-2 hibitors for managing β-thalassemia through computational 3 methods” by Pokharel et al, the authors. Unfortunately, the manuscript has major issues which preclude publication.
Major Points:
1) The authors make very basic mistakes in the manuscript suggesting that their understanding of the material is superficaial at best. Errors such as: "Likewise, F580 formed a stable H-bond with both TOP1 and TOP2 in 502 both 100 and 200 ns.” which is conflating times and equilibrium energies and Table 5 - „van der Waals” is corect, not „Vander wall”. A more thorough understanding of the underlying principles needs to be demonstrated by the authors before publication.
2) The authors use a very high cut-off with their tanimoto chemical similarity search. This finds highly similar molecules which are likely to be binders due to chemical similarity alone. Finding molecules that differ by a methyl group or less is not interesting or informative.
3) these results are entirely computational. Experimental confirmation of these results is needed for publication.
Minor Points:
-
Titles in tables 3 & 4 are not uniformly aligned with each other.
-
„Stability is mostly influenced by the formation of hydrogen bonds and hydrophobic interactions” (line 325) - this assertion needs to be supported by citation and numerical data produced by the calculations undertaken in this work. It does not hold for every protein ligand complex as often water, ionic interactions and long distance electrostatic effects outweigh the author's asserted interactions.
English needs significant help. While the grammar is usually passable, there are phrases that are unacceptable because of (hopefully) thesaurus errors. For example in the abstract: "this putative compound" is incorrect because the compound is a compound, its not probably a compound. The article is unpublishable from this perspective already as is, although this is fixable.
Reviewer 3 Report
Comments and Suggestions for Authors
Dear Editor thank you for the opportunity to rewiev a manuscript: „Discovery of selective protein arginine methyltransferase 5 inhibitors for managing β-thalassemia through computational methods“ The study employs a multi-step computational approach, including virtual screening, molecular docking, molecular dynamics simulations, and MM-PBSA analysis, to identify potential PRMT5 inhibitors. The study provides a detailed analysis of the molecular interactions between the identified compounds and PRMT5, including hydrogen bonding, hydrophobic interactions, and π-π stacking, enhancing the understanding of the binding mechanisms. The study validates the docking results using molecular dynamics simulations and MM-PBSA analysis, which provides further insight into the stability and binding affinity of the identified compounds. As a weakness I see lack of experimental validation because study relies solely on computational methods, and there is no experimental validation of the predicted PRMT5 inhibitors. Also, study comprehensively analyzes the binding interactions and stability of the identified compounds, it does not explore other aspects such as selectivity, off-target effects, or pharmacokinetic properties, which are important considerations in drug development. In my opinion manuscript is very interesting and suitible for publication.
Reviewer 4 Report
Comments and Suggestions for Authors
The manuscript titled "Discovery of Selective Protein Arginine Methyltransferase 5 Inhibitors for Managing β-Thalassemia through Computational Methods" is an exceptional contribution to the field of drug discovery and β-thalassemia management. The authors have adeptly employed computational methods to identify selective inhibitors for protein arginine methyltransferase 5 (PRMT5), demonstrating a sophisticated understanding computational biology. But it requires some additional revisions
1.As in the plagiarism report it is showing 23%. Authors are requested to reduce plagiarism less that 20 (Acceptable value).
2. At Page 1 line 22-29 must be changes.
3. Describe common ligand for arginine methyltransferase 5 inhibitors and its pharmacophoric feature required in figure format.
4. In this article, the authors proposed a computational method for analyzing the data. However, it would be advantageous to complement this approach by comparing the computational results with in vitro experiments. By incorporating experimental data, the study could provide more robust validation and enhance the reliability of the findings. Additionally, conducting in vitro experiments would allow for a more comprehensive understanding of the biological relevance and practical implications of the computational predictions.
5. Ravikumar Y, Koonyosying P, Srichairatanakool S, Ponpandian LN, Kumaravelu J, Srichairatanakool S. In Silico Molecular Docking and Dynamics Simulation Analysis of Potential Histone Lysine Methyl Transferase Inhibitors for Managing β-Thalassemia. Molecules. 2023 Oct 25;28(21):7266. most of the materials taken as it is and showing plagism. pls reconstruct your material and methods.
6. Overall the manuscript is well written, well managed and suitable for publication after revision.
Comments on the Quality of English Language
minor English editing required
Round 2
Reviewer 2 Report
Comments and Suggestions for Authors
While the use of language editing software has generally improved the language, there are still many, many language issues. Furthermore, the authors engage in overly optimistic, unfounded and unsupported (by either literature or experimental evidence) speculation. For example: ", oral bioavailability is a crucial parameter, and the low molecular weight, low TPSA, and total hydrogen bond donor and acceptors of the TOP1 and TOP2 compounds showed that they possess better oral bioavailability and good passive intestinal absorption" which is not supported nor do the number of hydrogen bond neccessarily correspond to intestinal absorption. Additionally: "It is believed that the compounds that obey Lipinski’s rule of five have better polarity and better folding and tend to exhibit good therapeutic effect", however the authors conflate protein folding with small molecule behavior here. Lipinski's rule defines solubility and other bio-availability behaviors, not therapuetic "goodness". Just because a compound is intestinally absorbed does not mean it will be an effective drug or even pharmacologically useful.
Comments on the Quality of English LanguageEnglish language editing still sorely needed.
Reviewer 4 Report
Comments and Suggestions for Authors
All comments have been incorporated into the manuscript. However, a major section regarding in vitro validation of the data presented is lacking. The authors are advised to include in vitro validation within this manuscript.
